# Cramming versus threading of long amphiphilic oligomers into a polyaromatic capsule

Masahiro Yamashina[1], Shunsuke Kusaba[1], Munetaka Akita[1], Takashi Kikuchi[2] & Michito Yoshizawa[1]

Oligo(ethylene oxide)s are known as widely useable yet not very interactive amphiphilic compounds. Here we report that the long amphiphilic oligomers are bound by a polyaromatic capsule in two different manners, depending on the chain length. For instance, the shorter pentamer is crammed into the isolated cavity of the capsule, whereas the longer decamer is threaded into the capsule to form a 1:1 host–guest complex in a pseudo-rotaxane fashion. These unusual bindings occur instantly, spontaneously, and quantitatively even in water at room temperature, with relatively high binding constants ($K_a > 10^6 \, M^{-1}$). Isothermal titration calorimetry (ITC) studies reveal that enthalpic stabilization is a dominant driving force for both of the complexations through multiple host–guest CH-$\pi$ and hydrogen-bonding interactions. Furthermore, long oligomers with an average molecular weight of 1000 Da (e.g., 22-mer) are also threaded into the capsules to give pseudo-rotaxane-shaped 2:1 host–guest complexes in water, selectively.

[1] Laboratory for Chemistry and Life Science, Institute of Innovative Research, Tokyo Institute of Technology, 4259 Nagatsuta, Midori-ku, Yokohama 226-8503, Japan. [2] Rigaku Corporation, 3-9-12 Matsubaracho, Akishima, Tokyo 196-8666, Japan. These authors contributed equally: Masahiro Yamashina, Shunsuke Kusaba. Correspondence and requests for materials should be addressed to M.Yo. (email: yoshizawa.m.ac@m.titech.ac.jp)

To bind or not to bind substrate molecules is ubiquitous in biological receptors and enzymes under aqueous ambient conditions. Such host–guest events depend strictly on the size and shape complementarity between the guest structure and the host cavity, surrounded by the protein assemblies, and on their mutual interactions[1,2]. Similar behavior is observed in artificial host–guest systems. For example, Rebek's hydrogen-bonded capsules take up one molecule of long-chain alkanes in a helical fashion in their cylindrical cavities[3–5]. One of Gibb's hydrophobic effect-driven capsules also encapsulates long alkanes in a helical or folded fashion in its spherical cavity[6–8]. However, as observed for other molecular capsules reported previously, longer alkanes as well as other compounds with volumes larger than the host cavities are hardly bound by the capsules due to the limited spaces available[9–15].

Oligo(ethylene oxide)s (OEOs) are very useful amphiphilic molecules and components that find broad application in materials chemistry and biochemistry, because of the high solubility in both organic solvents and water as well as the low biological toxicity and chemical reactivity[16–18]. However, owing to the weak interactions with aliphatic and small aromatic frameworks, the encapsulation of OEOs within molecular cages and capsules in solution has been seldom achieved[19], except for aliphatic cyclodextrin tubes[20,21]. Therefore, thermodynamic insights into the host–guest interactions remained obscure to date. We thus focused our attention on polyaromatic panels as promising frameworks to interact with OEO chains[22,23] and employed polyaromatic capsule **1** (Fig. 1b)[24,25] to investigate the detailed host–guest interactions and the generation of novel host–guest complexes in water. The coordination-driven capsule, formed quantitatively from two Pt(II) ions and four bent bispyridine ligands, provides a spherical hydrophobic cavity (~ 1 nm in diameter and 480 Å$^3$ in free volume) encircled by the eight anthracene panels. Owing to the 12 exterior hydrophilic groups, the polyaromatic framework can interact with guest molecules only from the inside. The isolated cavity of **1** has been shown to fully accommodate not only hydrophobic molecules (e.g., cyclophanes, pyrenes, AIBN, fullerene $C_{60}$, and $S_8$)[26–29] but also hydrophilic ones (e.g., sucrose and oligo(lactic acid)s)[30,31] in aqueous media. However, the binding of large and long molecules, particularly whose sizes and volumes are larger than the cavity, had not been achieved so far by the capsule.

In contrast to such common complexation behavior, herein we report that a molecular capsule binds one molecule of linear amphiphilic oligomers in two different manners, a cramming or threading fashion, depending on the chain length. Namely, the shorter oligomers are fully accommodated in the closed cavity, yet the longer ones stick the ends out from the cavity (Fig. 1a). These unusual host–guest events take place instantly, spontaneously, and quantitatively even in water at room temperature. In addition, detailed isothermal titration calorimetry (ITC) studies reveal that the two different structures are formed through enthalpy-driven complexation with high binding constants ( > 10$^6$ M$^{-1}$ for the 1:1 host–guest complexes). The obtained, threaded products are uncommon pseudo-rotaxane-shaped capsules, which would be a new promising building block for functional interlocked nanostructures such as molecular machines and soft materials[32–34].

## Results

**Cramming of short amphiphilic oligomers**. First, we found out that one molecule of methyl-capped OEOs **nEO** (n = 4–8; Fig. 1c and see Supplementary Fig. 1), with extended molecular lengths of up to ~ 3.2 nm, is quantitatively crammed into the cavity of polyaromatic capsule **1** in water. For example, when capsule **1**

(0.39 μmol) and pentamer **5EO** (0.39 μmol) were combined in $D_2O$ (0.5 mL) at room temperature, 1:1 host–guest complex **1•5EO** was exclusively formed within 5 min (Fig. 2a, right), which was confirmed by nuclear magnetic resonance spectroscopy (NMR), mass spectrometry (MS), and X-ray crystallographic analyses. The $^1H$ NMR spectrum of the resultant solution showed a sharp signal derived from the terminal methyl groups of **5EO** at –0.23 p.p.m. with an outstanding upfield shift ($\Delta\delta = -3.62$ p.p.m.) upon full encapsulation in the polyaromatic cavity of **1** (Fig. 2b, c and see Supplementary Fig. 2). Multiple methylene signals for **5EO** within **1** were also highly upfield shifted and observed in the range of –0.08–0.28 p.p.m. ($\Delta\delta_{max} = -3.80$ p.p.m.). The observed, sharp aromatic signals of **1** and the upfield-shifted proton $H_a$ indicate full encapsulation of the amphiphilic chain of **5EO** within **1** in water. Besides the NMR integration, a 1:1 host–guest ratio of the product was evidenced by the Electrospray ionization time-of-flight mass spectrometry (ESI-TOF MS) spectrum, where prominent peaks assignable to [**1•5EO** – $n•NO_3^-$]$^{n+}$ (n = 4–2) species were found (Fig. 2e and see Supplementary Fig. 3). Similarly, methyl-capped tetramer, hexamer, heptamer, and octamer were crammed into the isolated cavity of **1** to form the corresponding 1:1 host–guest complexes **1•nEO** (n = 4, 6–8) in a quantitative fashion (see Supplementary Figs. 4–12). It is worthy of note that the relative order for the host-binding affinity is **5EO ≈ 6EO > 7EO > 4EO >> 8EO** (430 Å$^3$ in volume), owing to their volume and shape complementarity, demonstrated by the $^1H$ NMR competitive binding experiments (see Supplementary Methods and Supplementary Figs. 13–16). The terminal methyl groups on OEOs are not essential for the formation of complexes **1•nEO**. For example, non-capped pentamer **5EO'** and octamer **8EO'** (Fig. 1c) were quantitatively encapsulated within **1** under the same conditions, respectively (see Supplementary Methods and Supplementary Figs. 17–20).

The X-ray crystal structure of host–guest complex **1•5EO** revealed that acyclic **5EO** is fully accommodated in the closed polyaromatic shell of capsule **1** (Fig. 3a, b and see Supplementary Figs. 21 and 22). Pale yellow single crystals of **1•5EO** were obtained by slow concentration of the saturated $H_2O$ solution at room temperature for 1 month. In the crystal structure, the bound **5EO** adopts a roughly coiled conformation in the cavity, in which the two methyl and four ethylene moieties of **5EO** are in close contact ( ≤ 3.8 Å) with the five anthracene panels of **1**, indicating the presence of effective host–guest CH-π interactions (Fig. 3c)[30,31]. In addition, six host–guest hydrogen-bonding interactions between the pyridine α-hydrogen atoms ($H_f$) and the ether oxygen atoms as well as three intramolecular hydrogen bonds of the guest were observed in the cavity[35,36]. The optimized structures of **1•nEO** (n = 6 and 8) also suggested that the spherical framework of **1** forces the flexible OEO chains into coiled conformations in the cavity (see Supplementary Fig. 23).

**Cramming of cyclic amphiphilic oligomers**. Next, we investigated the effect of the structural flexibility of short OEOs on the encapsulation within capsule **1** using acyclic **5EO** and the cyclic analog, 18-crown-6 (**CE**), by $^1H$ NMR and ITC analyses. In a manner similar to **1•5EO**, 1:1 host–guest complex **1•CE** formed quantitatively upon mixing **1** with **CE** in water at room temperature (Fig. 2a, left), in spite of the bulky cyclic structure (~ 0.9 nm in outer diameter). Upfield-shifted proton peaks for the encapsulated **CE** (Fig. 2d and see Supplementary Fig. 24) and prominent mass peaks for [**1•CE** – $n•NO_3^-$]$^{n+}$ (n = 4–2; see Supplementary Fig. 25) were observed in the $^1H$ MNR and ESI-TOF MS spectra, respectively. The coordination bond between the Pt(II) ion and the bispyridine ligand is inert at ambient temperature so that the encapsulation of the linear as well as

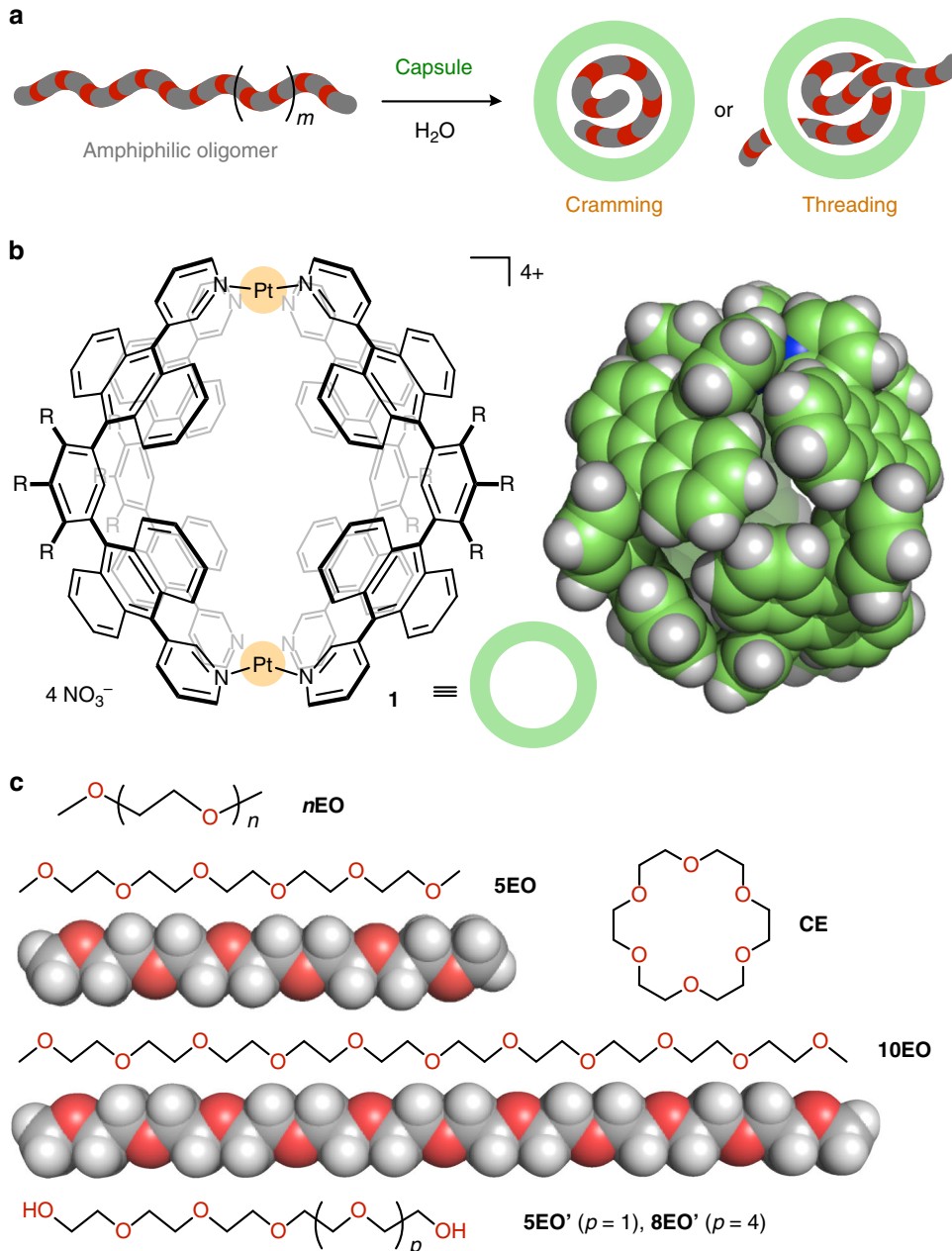

**Fig. 1** Design and components for the formation of host–guest complexes with cramming or threading amphiphilic oligomers. **a** Schematic representation of the cramming or threading of long amphiphilic oligomers into a molecular capsule. **b** Polyaromatic capsule **1** (R = -OCH$_2$CH$_2$OCH$_3$) and the X-ray crystal structure (the peripheral substituents are replaced by hydrogen atoms for clarity). **c** Methyl-capped oligo(ethylene oxide)s (*n*EO) and its pentamer **5EO** and decamer **10EO** with their optimized structures. Cyclic pentamer **CE** and non-capped pentamer **5EO′** and octamer **8EO′**

cyclic oligomers by **1** should occur without any dissociation step of the capsular frameworks. A $^1$H NMR competitive binding experiment of capsule **1** with **5EO** and **CE** (1.0 equiv. each) in water resulted in the quantitative formation of a mixture of **1•5EO** (90%) and **1•CE** (10%) at room temperature (see Supplementary Fig. 26). The products' yields were altered to 25% and 75%, respectively, after the mixture was heated at 60 °C for 9 h. This finding can be interpreted in terms of flexible **5EO** being kinetically trapped by **1** in preference to rigid **CE** owing to the narrow space between the polyaromatic frameworks.

**Thermodynamic parameters for the formation of 1•5EO and 1•CE.** In order to obtain thermodynamic insights into the

formation of **1•5EO** and **1•CE**, the ITC studies were conducted in H$_2$O at 25 °C (Fig. 4a, b and see Supplementary Figs. 27 and 28). Negative, large enthalpy and small entropy changes ($\Delta H = -52.8$ kJ mol$^{-1}$ and T$\Delta S = -16.6$ kJ mol$^{-1}$) were measured for **1•5EO** (Table 1 and see Supplementary Table 1), indicating that the complexation is driven by enthalpic stabilization. Interestingly, the enthalpic contribution is larger than that for **1•CE** ($\Delta\Delta H = -4.9$ kJ mol$^{-1}$) but the entropic contribution is smaller than that for **1•CE** (T$\Delta\Delta S = -5.0$ kJ mol$^{-1}$), owing to the flexible structure of acyclic **5EO**. The large enthalpic gain most probably arises from multiple host–guest CH-π and hydrogen-bonding interactions by cramming of the guest into the polyaromatic host cavity (Fig. 3a–c), in combination with the release of bound high-energy water clusters from the capsule[37–39]. The binding constant

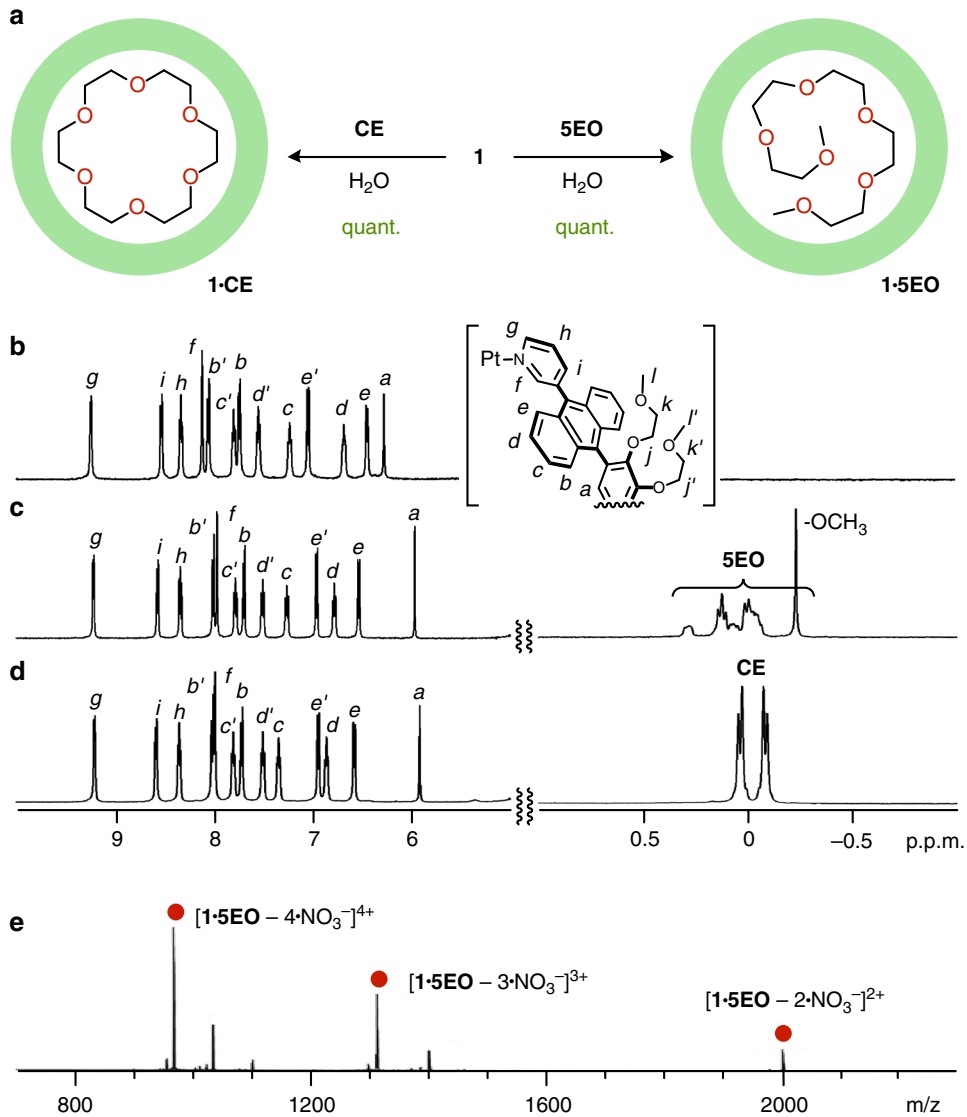

**Fig. 2** Formation and characterization of host–guest complexes **1•5EO** and **1•CE** in a cramming fashion. **a** Schematic representation of the formation of 1:1 host–guest complexes **1•5EO** and **1•CE** in water. $^1$H NMR spectra (500 MHz, $D_2O$, room temperature) of **b 1**, **c 1•5EO**, and **d 1•CE**. **e** ESI-TOF MS spectrum ($H_2O$, room temperature) of **1•5EO**

for **1•5EO** ($K_a = 2.1 \times 10^6\,M^{-1}$) is relatively high even in water and comparable to that for **1•CE** (Table 1 and see Supplementary Table 1). The optimized structure of **1•CE** indicates that cyclic **CE** adopts a bent conformation within **1** to fit the spherical cavity (Fig. 3d and see Supplementary Fig. 23).

**Threading of long amphiphilic oligomers and their thermo-dynamic parameters**. Unexpectedly, longer OEOs ($n \geq 10$) were bound by capsule **1** to form pseudo-rotaxane-shaped host–guest complexes in water. For example, methyl-capped decamer **10EO** is composed of a long amphiphilic chain with an extended length of ~4 nm and a volume of ~ 520 Å$^3$ (Fig. 1c), which is roughly 1.1-times larger than the free volume of the cavity of capsule **1**. Nevertheless, mixing **1** with **10EO** (1.0 equiv.) in water at room temperature led to the quantitative formation of 1:1 host–guest complex **1•10EO** within 5 min (Fig. 5a). The product is stable enough under standard ESI-TOF MS conditions so that the host–guest ratio could be unambiguously confirmed by the MS

analysis: prominent molecular ion peaks were observed at $m/z$ 2113.4, 1388.3, and 1025.7 for [**1•10EO** – $n$•$NO_3^-$]$^{n+}$ ($n$ = 2, 3, and 4, respectively; Fig. 5b and see Supplementary Fig. 29). However, in sharp contrast to **1•5EO**, the $^1$H NMR spectrum of **1•10EO** displayed very broad peaks for **1** and bound **10EO** in the aromatic and highly upfield (~ 0.5 p.p.m.) regions, respectively, at room temperature (Fig. 5c and see Supplementary Fig. 30). Large upfield shifts of the guest ethylene signals indicate the incorporation of the middle part rather than the terminal part of the guest chain into the polyaromatic cavity (Fig. 5a, upper right and lower left, respectively). The terminal methyl signals could not be identified even at various temperatures (see Supplementary Fig. 31), most probably owing to the restricted motion of the chain on the NMR timescale. Upon cooling to 5 °C, the broadened aromatic signals of capsule **1** sharpened to be ~ 24 peaks including two singlet signals derived from the inner protons $H_a$ (Fig. 5d). The signal pattern indicates the desymmetrization of the spherical framework of **1** with a virtual $D_{4h}$ symmetry upon

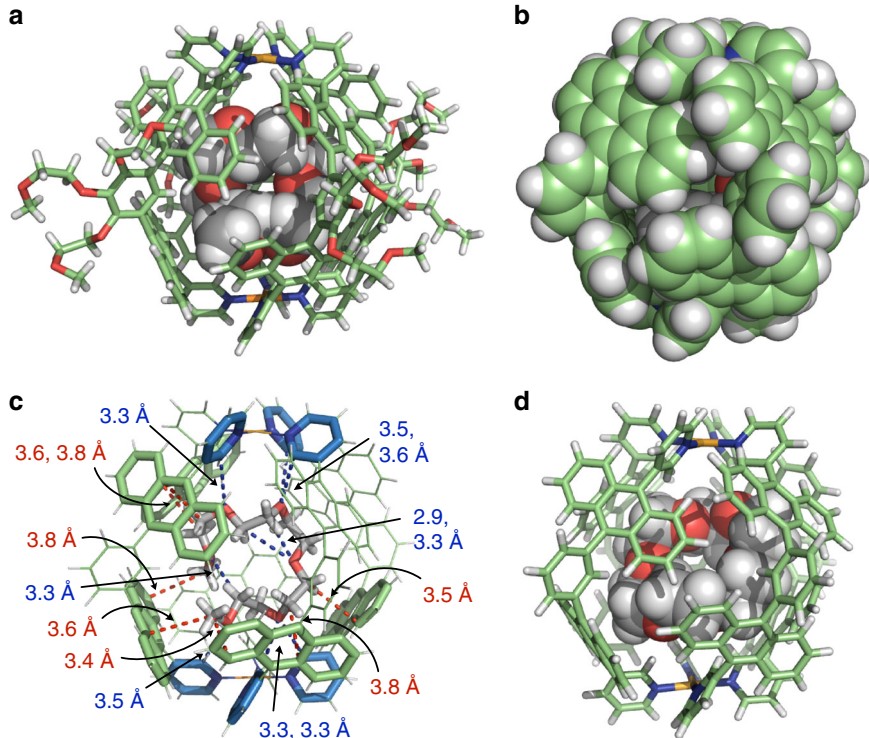

**Fig. 3** Crystal and optimized structures of host–guest complexes **1•5EO** and **1•CE**, respectively. **a** X-ray crystal structure of **1•5EO** and **b** its space-filling representation (the peripheral substituents of **1** are replaced by hydrogen atoms for clarity). **c** Highlighted host–guest and guest-guest interactions of **1•5EO** in the cavity (red and blue dashed lines are CH-π and hydrogen-bonding interactions, respectively). **d** Optimized structure of **1•CE** (the peripheral substituents of **1** are replaced by hydrogen atoms for clarity)

binding of **10EO**. Molecular modeling studies combined with the NMR and MS findings suggest that the obtained host–guest complex **1•10EO** adopts a pseudo-rotaxane structure, in which the long chain of **10EO** penetrates the polyaromatic shell of **1** through the diagonal, small slits between the polyaromatic panels (Fig. 5e, f and see Supplementary Fig. 23). In contrast, one side of a long **9EO** chain might partially stick out of the capsular shell. 1:1 Host–guest complex **1•9EO** displayed both thoroughly broad signals and relatively sharp signals for the bound nonamer in the $^1$H NMR spectrum (see Supplementary Methods and Supplementary Figs. 32–35).

Thermodynamic parameters for the formation of **1•10EO** obtained by the ITC analysis (Fig. 4c, Table 1 and see Supplementary Fig. 36 and Table 1) demonstrate that the 1:1 host–guest complex formation in water is also driven by large enthalpic stabilization ($\Delta H = -59.5$ kJ mol$^{-1}$). The optimized structure of **1•10EO** also suggests the existence of multiple host–guest CH-π and hydrogen-bonding interactions in the cavity and at the openings of **1** (Fig. 5e). The large negative entropy change ($T\Delta S = -25.1$ kJ mol$^{-1}$) as compared with that for **1•5EO** ($-16.6$ kJ mol$^{-1}$) is most likely derived from the large restriction of the molecular motion of long **10EO** threaded through **1**. The binding constant of **1** for **10EO** is still relatively high ($K_a = 1.1 \times 10^6$ M$^{-1}$), which is approximately half of those for **5EO** and **CE**.

**Formation of 2:1 host–guest complexes and their thermodynamic parameters**. To investigate the synthetic scope of pseudo-rotaxane-shaped host–guest complexes, we employed methyl-capped, long OEOs **OEO$^{1000}$** and **OEO$^{2000}$** with average molecular weights of 1000 and 2000 Da, respectively. Oligomers

**OEO$^{1000}$** and **OEO$^{2000}$** contain a mixture of CH$_3$O(CH$_2$CH$_2$O)$_n$CH$_3$ ($n_{\text{average}} = \sim 22$ and $\sim 44$, respectively). When capsule **1** (0.39 μmol) and **OEO$^{1000}$** (0.19 μmol) were mixed in water (0.5 mL) at room temperature, to our surprise, 2:1 host–guest complexes **(1)$_2$•OEO$^{1000}$** were predominantly generated within 5 min (Fig. 6a). The room temperature $^1$H NMR spectrum of **(1)$_2$•OEO$^{1000}$** showed relatively broadened peaks derived from the threaded **OEO$^{1000}$** as well as sharp and split peaks derived from the polyaromatic framework of **1** (Fig. 6b and see Supplementary Figs. 37 and 38). The proton NMR signal pattern of **(1)$_2$•OEO$^{1000}$** closely resembles that of 1:1 pseudo-rotaxane complex **1•10EO** at 5 °C (Fig. 5d and see Supplementary Fig. 31). The broadening of all of the proton signals for bound **OEO$^{1000}$** is construed as the shuttling motion of the capsules along the long chain (~7 nm in extended length) on the NMR timescale at room temperature. The ESI-TOF MS spectrum showed molecular ion peaks assignable to a 1:1 host–guest complex (see Supplementary Fig. 39), probably due to enhanced cationic repulsion under the MS conditions. However, the 2:1 host–guest ratio suggested by the $^1$H NMR titration experiments (see Supplementary Fig. 40) was unequivocally evidenced by the ITC studies (Fig. 6c and see Supplementary Fig. 41). Regarding to the complex formation, the enthalpy change for **(1)$_2$•OEO$^{1000}$** was estimated to be a large negative value ($\Delta H = -131.3$ kJ mol$^{-1}$), which is approximately twice of that for **1•10EO** (Table 1 and see Supplementary Table 1). The entropy change for **(1)$_2$•OEO$^{1000}$** ($T\Delta S = -99.6$ kJ mol$^{-1}$) is approximately four times smaller than that for **1•10EO**. The large entropy loss is most probably caused by the restricted motion of the long and flexible chains of **OEO$^{1000}$** upon their threading into the polyaromatic cavities. The optimized structure of **(1)$_2$•OEO$^{1000}$**

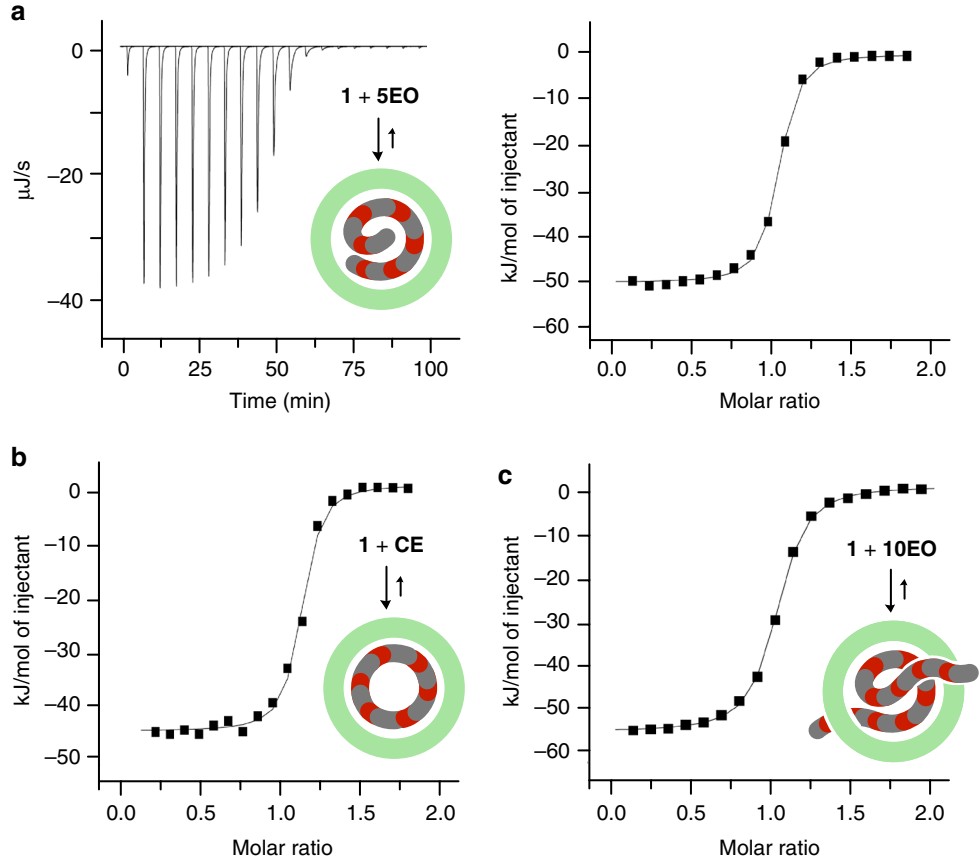

**Fig. 4** ITC data for the formation of host–guest complexes **1•5EO**, **1•CE**, and **1•10EO**. **a** ITC thermogram and binding isotherm ($H_2O$, 298 K) of **5EO** toward capsule **1**. ITC binding isotherms ($H_2O$, 298 K) of **b CE** and **c 10EO** toward **1**

### Table 1 Thermodynamic parameters and binding constants

| Entry | $\Delta H$ (kJ mol$^{-1}$) | $T\Delta S$ (kJ mol$^{-1}$) | $\Delta G$ (kJ mol$^{-1}$) | $K_a$ |
|---|---|---|---|---|
| **1•5EO** | − 52.76 ± 0.32 | − 16.58 | − 36.18 | (2.13 ± 0.22) × 10$^6$ M$^{-1}$ |
| **1•CE** | − 47.91 ± 0.51 | − 11.60 | − 36.31 | (2.29 ± 0.43) × 10$^6$ M$^{-1}$ |
| **1•10EO** | − 59.50 ± 0.18 | − 25.06 | − 34.44 | (1.06 ± 0.04) × 10$^6$ M$^{-1}$ |
| **(1)$_2$•OEO$^{1000}$** | − 131.29 ± 0.34 | − 99.62 | − 31.67 | (3.39 ± 0.06) × 10$^5$ M$^{-2}$ |

Thermodynamic parameters ($\Delta H$ and T$\Delta S$) and binding constants ($K_a$) for the formation of **1•5EO**, **1•CE**, **1•10EO**, and **(1)$_2$•OEO$^{1000}$** obtained by ITC experiments ($H_2O$, 298 K)

exhibits that the amphiphilic chain of **OEO$^{1000}$** ($q = 1$) is long enough to penetrate two molecules of capsule **1** separately (Fig. 6d).

Moreover, $^1$H NMR and ITC analyses revealed that similar 2:1 pseudo-rotaxane complexes **(1)$_2$•OEO$^{2000}$** were selectively formed upon simple mixing of **1** and **OEO$^{2000}$** in water (Fig. 6a and see Supplementary Figs. 42 and 43). The formation of **(1)$_2$•OEO$^{2000}$** is more favorable than that of **(1)$_2$•OEO$^{1000}$** on the basis of the estimated Gibbs free energy ($\Delta G$) and binding constant ($K_a$) by the ITC studies (Table 1 and see Supplementary Fig. 44 and Table 1). The observed, unusual regulation of the number of the threaded capsules by the long chains (up to ~ 44 mer) can be explained by the electrostatic repulsion between the tetravalent capsules[40], besides the energetic balance between the enthalpic gains and the entropic losses. To the best of our knowledge, this is the first example of a molecular capsule usable as a component for facile pseudo-oligorotaxane syntheses[32–34].

## Discussion

We have disclosed unusual host–guest complexation between a molecular capsule and long amphiphilic oligomers, i.e., oligo (ethylene oxide)s, in water at room temperature. The isolated polyaromatic cavity of the capsule quantitatively binds one molecule of the tetramer to the octamer in a cramming fashion. On the other hand, the longer oligomers such as the decamer and docosamer (22-mer) are quantitatively bound by the capsule in a threading fashion. The observed, straightforward, and chain length-dependent complexation is driven by effective enthalpic stabilization through multiple host–guest CH-π and hydrogen-bonding interactions in the cavity of the capsule. Although both oligo(ethylene oxide) chains and polyaromatic panels are well-studied and have been applied as building blocks for wide-ranging functional molecules, we, for the first time, revealed effective intermolecular interactions between the chains and the panels, which could prove to be useful chemical tools to construct

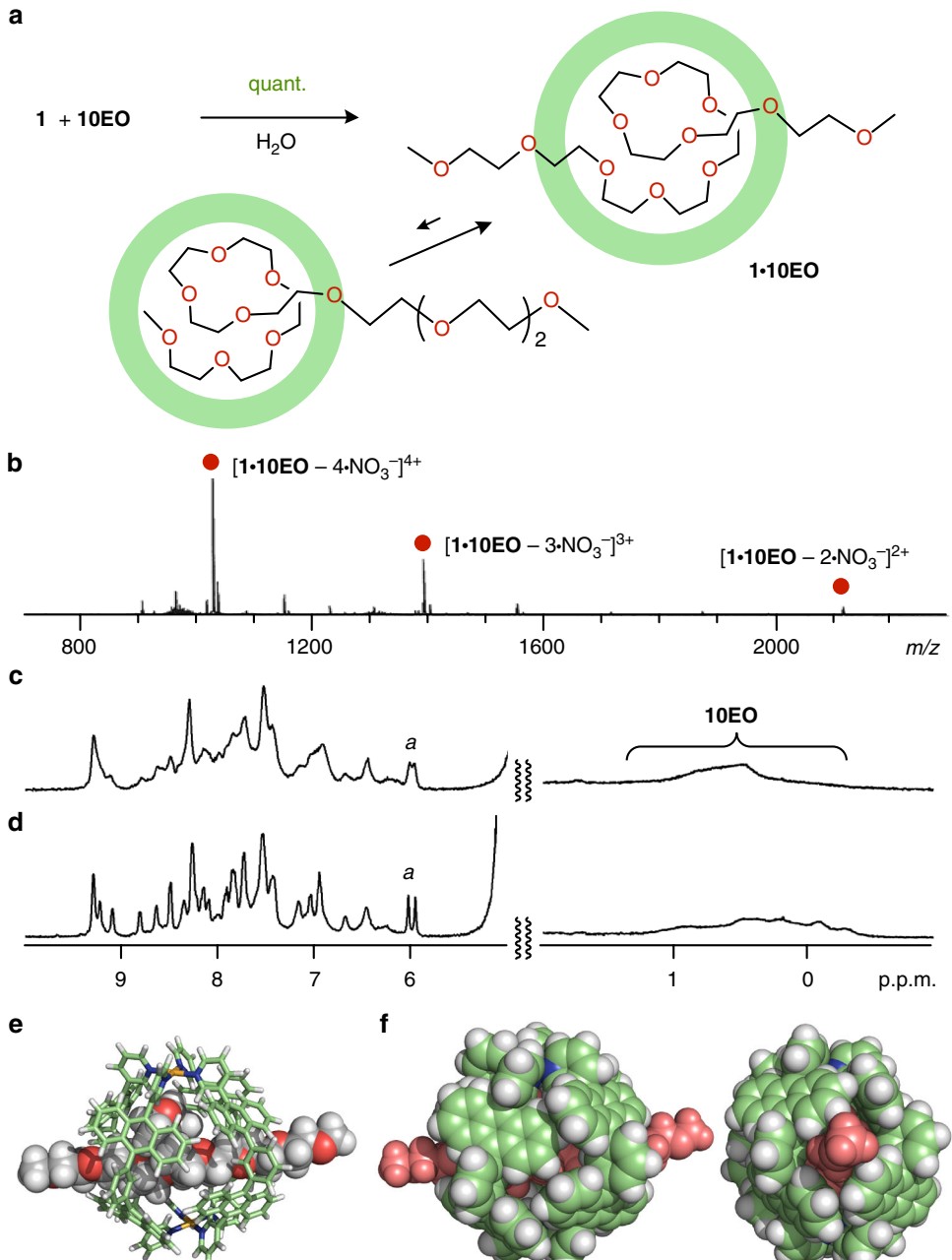

**Fig. 5** Formation and characterization of host–guest complex **1•10EO** in a threading fashion. **a** Schematic representation of the formation of 1:1 pseudorotaxane complex **1•10EO** and its equilibrium mixture in water. **b** ESI-TOF MS spectrum (H₂O, room temperature) of **1•10EO**. ¹H NMR spectra (500 MHz, D₂O) of **1•10EO** at **c** room temperature and **d** 5 °C. Optimized structure of **1•10EO**: **e** cylinder and space-filling representation and **f** space-filling representation highlighting threaded **10EO** in red (the peripheral substituents are replaced by hydrogen atoms for clarity)

new interlocked supramolecular structures and materials with controllable functions.

## Methods

**General**. NMR: Bruker AVANCE-400 (400 MHz) & ASCEND-500 (500 MHz), ESI-TOF MS: Bruker micrOTOF II, Single crystal XRD: Rigaku XtaLAB Pro MM007 HyPix-6000HE, ITC: MicroCal system, VP-ITC model, Theoretical calculation: Fujitsu Limited SCIGRESS program (version FJ 2.6). Solvents and reagents: TCI Co., Ltd., Wako Pure Chemical Industries Ltd., Kanto Chemical Co., Inc., Sigma-Aldrich Co., and Cambridge Isotope Laboratories, Inc. Ultra-pure water (Milli-Q) was used for the ITC analysis. Capsule **1** and methyl-capped oligo (ethylene oxide)s **5EO**-**10EO** were synthesized according to previously reported procedures[27,41]. The cavity volume estimated from the crystal structure of **1** (see Supplementary Fig. 21) was calculated with the PLATON program[42]. Probe radius:

1.4 Å, grid step: 0.2 Å, atomic radii: $C = 1.70$ Å; $H = 1.20$ Å; $O = 1.52$ Å; $N = 1.55$ Å; $Pt = 1.72$ Å.

**Formation of host–guest complexes 1•5EO and 1•5EO′**. Capsule **1** (1.5 mg, 0.39 μmol), **5EO** (0.10 mg, 0.39 μmol), and D₂O (0.5 ml) were added to a glass test tube. The mixture was stirred at room temperature (or 60 °C) for 5 min. The quantitative formation of **1•5EO** was confirmed by NMR, ESI-TOF MS, and ITC analyses (see Supplementary Figs. 2, 3, and 27). Under the same conditions, the treatment of **1** with **5EO′** quantitatively afforded **1•5EO′** in water (see Supplementary Figs. 17 and 18). The proton signals of free **5EO** were observed in the range of 3.65 to 3.72 p.p. m. upon addition of excess **5EO** to **1** in D₂O.

**1•5EO**: ¹H NMR (500 MHz, D₂O, room temperature): δ −0.23 (s, 6H, **5EO**), −0.08–0.18 (m, 18H, **5EO**), 0.28 (m, 2H, **5EO**), 2.49 (s, 24H, **1**), 3.13 (m, 16H, **1**), 3.49 (s, 12H, **1**), 3.95 (t, *J* = 4.0 Hz, 8H, **1**), 3.99 (m, 8H, **1**), 4.09 (m, 8H, **1**), 4.49 (m, 4H, **1**), 4.63 (m, 4H, **1**), 5.98 (s, 4H, **1**), 6.54 (d, *J* = 9.0 Hz, 8H, **1**), 6.79 (dd, *J* = 9.0,

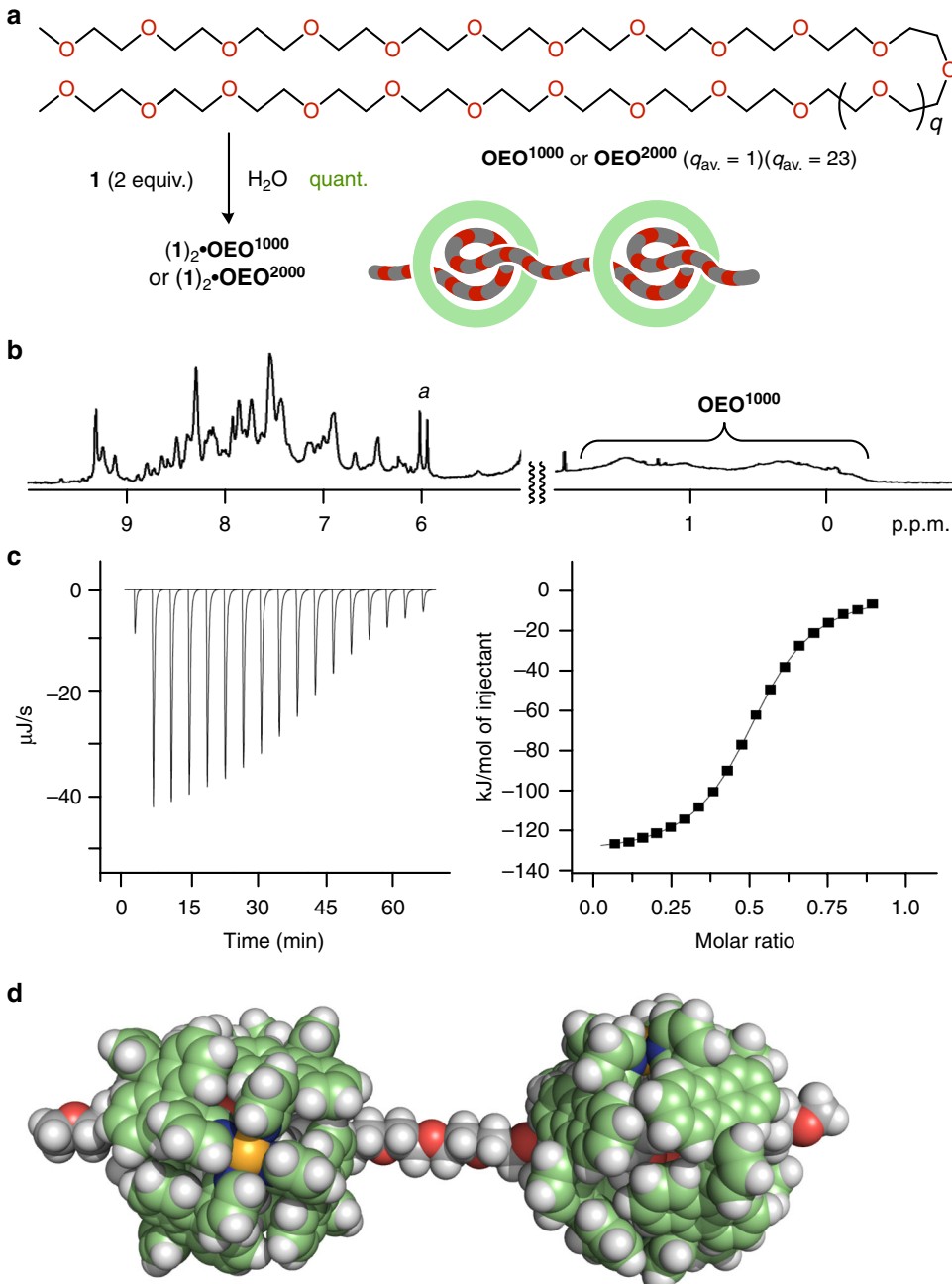

**Fig. 6** Formation and characterization of 2:1 host–guest complexes (**1**)₂•**OEO**^**1000** and (**1**)₂•**OEO**^**2000** in a threading fashion. **a** Schematic representation of the formation of pseudo-rotaxane-shaped 2:1 host–guest complexes (**1**)₂•**OEO**^**1000** and (**1**)₂•**OEO**^**2000** in water. **b** ¹H NMR spectrum (500 MHz, D₂O, room temperature) of (**1**)₂•**OEO**^**1000**. **c** ITC thermogram and binding isotherm (H₂O, 298 K) of **OEO**^**1000** toward capsule **1**. **d** Optimized structure of (**1**)₂•**OEO**^**1000** ($q = 1$) (the peripheral substituents are replaced by hydrogen atoms for clarity)

7.5 Hz, 8H, **1**), 6.97 (d, $J = 9.0$ Hz, 8H, **1**, 7.27 (dd, $J = 9.0$, 7.5 Hz, 8H, **1**, 7.51 (dd, $J = 9.0$, 7.5 Hz, 8H, **1**), 7.70 (d, $J = 9.0$ Hz, 8H, **1**), 7.79 (dd, $J = 9.0$, 7.5 Hz, 8H, **1**), 7.97 (s, 8H, **1**), 8.01 (d, $J = 9.0$ Hz, 8H, **1**), 8.34 (dd, $J = 8.0$, 5.5 Hz, 8H, **1**), 8.57 (d, $J = 8.0$ Hz, 8H, **1**), 9.22 (d, $J = 5.5$ Hz, 8H, **1**). ESI-TOF MS (H₂O): $m/z$ 2003.7 [**1•5EO** − 2•NO₃⁻]²⁺, 1315.2 [**1•5EO** − 3•NO₃⁻]³⁺, 970.9 [**1•5EO** − 4•NO₃⁻]⁴⁺.

**1•5EO':** ¹H NMR (500 MHz, D₂O, room temperature): δ −0.36 (br, 2H, **5EO'**), −0.14 (br, 2H, **5EO'**), −0.02–0.15 (m, 14H, **5EO'**), 0.21 (br, 2H, **5EO'**), 2.48 (s, 24H, **1**), 3.12 (m, 16H, **1**), 3.48 (s, 12H, **1**), 3.95 (t, $J = 4.0$ Hz, 8H, **1**), 3.99 (m, 8H, **1**), 4.08 (m, 8H, **1**), 4.48 (m, 4H, **1**), 4.62 (m, 4H, **1**), 6.01 (s, 4H, **1**), 6.47 (d, $J = 9.0$ Hz, 8H, **1**), 6.73 (dd, $J = 9.0$, 7.5 Hz, 8H, **1**), 6.96 (d, $J = 9.0$ Hz, 8H, **1**), 7.25 (dd, $J = 9.0$, 7.5 Hz, 8H, **1**), 7.51 (dd, $J = 9.0$, 7.5 Hz, 8H, **1**), 7.70 (d, $J = 9.0$ Hz, 8H, **1**), 7.78 (dd, $J = 9.0$, 7.5 Hz, 8H, **1**), 8.01 (s, 8H, **1**), 8.02 (d, $J = 9.0$ Hz, 8H, **1**), 8.33 (dd, $J = 7.5$, 5.5 Hz, 8H, **1**), 8.54 (d, $J = 7.5$ Hz, 8H, **1**), 9.22 (d, $J = 5.5$ Hz, 8H, **1**). ESI-TOF MS (H₂O): $m/z$ 1989.8 [**1•5EO'** − 2•NO₃⁻]²⁺, 1305.9 [**1•5EO'** − 3•NO₃⁻]³⁺, 963.9 [**1•5EO'** − 4•NO₃⁻]⁴⁺.

**Formation of host–guest complex 1•CE.** Capsule **1** (1.5 mg, 0.39 μmol), 18-crown-6 (**CE**; 0.10 mg, 0.39 μmol), and D₂O (0.5 ml) were added to a glass test tube. The mixture was stirred at room temperature (or 60 °C) for 5 min. The quantitative formation of **1•CE** was confirmed by NMR, ESI-TOF MS, and ITC analyses (see Supplementary Figs. 24, 25, and 28).

¹H NMR (500 MHz, D₂O, room temperature): δ −0.09 (d, $J = 8.7$ Hz, 12H, **CE**), 0.03 (d, $J = 8.7$ Hz, 12H, **CE**), 2.53 (s, 24H, **1**), 3.19 (m, 16H, **1**), 3.51 (s, 12H, **1**), 3.97 (m, 8H, **1**), 4.06 (m, 8H, **1**), 4.13 (m, 8H, **1**), 4.52 (m, 4H, **1**), 4.64 (m, 4H, **1**), 5.92 (s, 4H, **1**), 6.58 (d, $J = 9.0$ Hz, 8H, **1**), 6.86 (dd, $J = 9.0$, 7.5 Hz, 8H, **1**), 6.94 (d, $J = 9.0$ Hz, 8H, **1**), 7.35 (dd, $J = 9.0$, 7.5 Hz, 8H, **1**), 7.50 (dd, $J = 9.0$, 7.5 Hz, 8H, **1**), 7.72 (d, $J = 9.0$ Hz, 8H, **1**), 7.80 (dd, $J = 9.0$, 7.5 Hz, 8H, **1**), 7.98 (s, 8H, **1**), 8.01 (d, $J = 9.0$ Hz, 8H, **1**), 8.34 (dd, $J = 7.5$, 5.5 Hz, 8H, **1**), 8.58 (d, $J = 7.5$ Hz, 8H, **1**), 9.20 (d, $J = 5.5$ Hz, 8H, **1**). ESI-TOF MS (H₂O): $m/z$ 2002.6 [**1•CE** − 2•NO₃⁻]²⁺, 1314.4 [**1•CE** − 3•NO₃⁻]³⁺, 970.3 [**1•CE** − 4•NO₃⁻]⁴⁺.

**Formation of host–guest complex 1•10EO.** Capsule **1** (1.5 mg, 0.39 μmol), **10EO** (0.20 mg, 0.39 μmol), and $D_2O$ (0.5 ml) were added to a glass test tube. The mixture was stirred at room temperature (or 60 °C) for 5 min. The quantitative formation of **1•10EO** was confirmed by NMR, ESI-TOF MS, and ITC analyses (see Supplementary Figs. 29, 30, and 36).

$^1$H NMR (500 MHz, $D_2O$, room temperature): δ −0.52–1.80 (br, 30H, **10EO**), 2.15–2.67 (m, 24H, **1**), 2.96–3.30 (m, 16H, **1**), 3.50 (s, 12H, **1**), 3.96 (br, 8H, **1**), 4.07 (br, 16H, **1**), 4.52 (br, 4H, **1**), 4.62 (br, 4H, **1**), 5.96 (s, 2H, **1**), 6.01 (s, 2H, **1**), 6.09–7.25 (m, 26H, **1**), 7.25–8.89 (m, 64H, **1**), 9.01–9.39 (m, 6H, **1**). ESI-TOF MS ($H_2O$): $m/z$ 2113.4 [**1•10EO** − 2•$NO_3^-$]$^{2+}$, 1388.3 [**1•10EO** − 3•$NO_3^-$]$^{3+}$, 1025.7 [**1•10EO** − 4•$NO_3^-$]$^{4+}$.

**Formation of host–guest complexes (1)$_2$•OEO$^{1000}$ and (1)$_2$•OEO$^{2000}$.** Capsule **1** (1.5 mg, 0.39 μmol), **OEO$^{1000}$** (0.19 mg, 0.19 μmol), and $D_2O$ (0.5 ml) were added to a glass test tube. The mixture was stirred at room temperature for 5 min. The quantitative formation of **(1)$_2$•OEO$^{1000}$** was confirmed by NMR and ITC analyses (see Supplementary Figs. 37 and 41). Under the same conditions, the treatment of **1** with **OEO$^{2000}$** quantitatively gave rise to **(1)$_2$•OEO$^{2000}$** in water (see Supplementary Figs. 42 and 44).

**(1)$_2$•OEO$^{1000}$:** $^1$H NMR (500 MHz, $D_2O$, room temperature): δ −0.48–1.89 (br, 30H, **OEO$^{1000}$**), 2.16–2.68 (m, 24H, **1**), 2.86–3.31 (m, 16H, **1**), 3.51 (br, 12H, **1**), 3.82–4.27 (m, 24H, **1**), 4.52 (br, 4H, **1**), 4.62 (br, 4H, **1**), 5.95 (s, 2H, **1**), 6.02 (s, 2H, **1**), 6.10–7.25 (m, 26H, **1**), 7.25–8.94 (m, 64H, **1**), 8.99–9.41 (m, 6H, **1**).

**(1)$_2$•OEO$^{2000}$:** $^1$H NMR (500 MHz, $D_2O$, room temperature): δ −0.34–1.84 (br, 30H, **OEO$^{2000}$**), 2.17–2.69 (m, 24H, **1**), 2.89–3.29 (m, 16H, **1**), 3.48 (br, 12H, **1**), 3.81–4.25 (m, 24H, **1**), 4.49 (br, 4H, **1**), 4.62 (br, 4H, **1**), 5.94 (s, 2H, **1**), 6.00 (s, 2H, **1**), 6.08–6.57 (m, 8H, **1**), 6.57–7.25 (m, 18H, **1**), 7.25–8.86 (m, 64H, **1**), 9.01–9.44 (m, 6H, **1**).

**ITC analysis for the complex formation of 1 with 5EO, CE, 10EO, OEO$^{1000}$, and OEO$^{2000}$.** ITC measurements were performed by dropping the $H_2O$ solutions (1.77–3.78 mM) of **5EO**, **CE**, **10EO**, **OEO$^{1000}$**, or **OEO$^{2000}$** to the $H_2O$ solutions (0.11–0.14 mM) of capsule **1** at 25 °C (see Supplementary Figs. 27, 28, 36, 41, and 44 and Table 1).

**X-ray crystal data of 1.** $C_{212}H_{184}N_8O_{24}Pt_2$, $M_r = 3617.84$, Triclinic, $P$-1, $a = 21.2847(5)$ Å, $b = 21.4245(5)$ Å, $c = 27.3518(5)$ Å, $V = 10003.1(4)$ Å$^3$, $Z = 2$, $\rho_{calcd} = 1.201$ g cm$^{-3}$, $F(000) = 3720.0$, $T = 93$ K, reflections collected/unique 80903/34070 ($R_{int} = 0.0393$), $R_1 = 0.0834$ ($I > 2\sigma(I)$), $wR_2 = 0.2714$, GOF 1.049. All the diffraction data were collected on a Rigaku XtaLAB Pro MM007 HyPix-6000HE diffractometer ($\lambda$(CuK$\alpha$) = 1.54184 Å). The contribution of the electron density associated with greatly disordered counterions and solvent molecules, which could not be modeled with discrete atomic positions, were handled using the solvent mask in the Olex$^2$ program (see Supplementary Methods and Supplementary Table 2).

**X-ray crystal data of 1•5EO.** $C_{216}H_{192.66}N_8O_{25.5}Pt_2$, $M_r = 3698.61$, Triclinic, $P$-1, $a = 19.4231(13)$ Å, $b = 20.4314(14)$ Å, $c = 28.8216(18)$ Å, $V = 9820.3(12)$ Å$^3$, $Z = 2$, $\rho_{calcd} = 1.251$ g cm$^{-3}$, $F(000) = 3809.0$, $T = 93$ K, reflections collected/unique 81517/20534 ($R_{int} = 0.1983$), $R_1 = 0.1273$ ($I > 2\sigma(I)$), $wR_2 = 0.4023$, GOF 1.017. All the diffraction data were collected on a Rigaku XtaLAB Pro MM007 HyPix-6000HE diffractometer ($\lambda$(CuK$\alpha$) = 1.54184 Å). The contribution of the electron density associated with greatly disordered counterions and solvent molecules, which could not be modeled with discrete atomic positions, were handled using the solvent mask in the Olex$^2$ program (see Supplementary Methods and Supplementary Table 3).

## Data availability

The authors declare that the data supporting the findings of this study are available within the Supplementary Information files and from the corresponding author upon reasonable request. CCDC 1840603 and CCDC 1842338 contain the supplementary crystallographic data for the structure reported in this article. The data can be obtained free of charge from The Cambridge Crystallographic Data Centre (CCDC) via www.ccdc.cam.ac.uk/data_request/cif.

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

## Acknowledgements

This work was supported by JSPS KAKENHI (Grant No. JP25104011/JP17H05359/JP18H01990) and Support for Tokyotech Advanced Researchers (STAR). We thank Dr. Takane Imaoka and Professor Kimihisa Yamamoto (Tokyo Institute of Technology) for assistance with ITC experiments. M.Ya. thanks the JSPS for an Overseas Research Fellowship.

## Author contributions

M.Ya., S.K., and M.Yo. designed the work, carried out research, analyzed data, and wrote the paper. M.A. was involved in the work discussion. T.K. contributed to X-ray crystallographic analysis. M.Yo. is the principal investigator. All authors discussed the results and commented on the manuscript.

## Additional information

**Competing interests:** The authors declare no competing interests.

