## [Peer Review File · Nature Communications]

Reviewers' comments:

Reviewer #1 (Remarks to the Author):

This is a very interesting and competently written manuscript that I recommend for publication after minor revision.

The formation of threaded capsules on ethylenoxide-oligomers is truly fascinating and the large high field shifts in the NMR, the ITC, MS and XRD evidence is overwhelming.

What should be included in a revised version, is the following:

- Are the counterions (nitrates) also within the capsule, or outside...? In the XRD in the SI, you omit those. From the MS, you say, the nitrates are in part included with the main signals, however, it is not clear to me if inside or outside the capsule.
- What is the effect on encapsulation on the ¹⁹⁵Pt-NMR...? Is there any difference...?
- In both crystal structures you do have A- and B-alerts in the CIFcheck files. However, you do not give any explanation for them. In addition, the second structure does not qualify for regular uptake in the CCDC, as R1 and wR2 values are high (R1 > 12%). I am familiar with such type of structures, so I do believe, the structures may be acceptable for publication.

However, at least every effort should be made to prove that the current state is as good as it gets. Please include a competent statement in the supporting information. I realize, that the thermal ellipsoids of the dangling residues are large (as expected). You even state that solvent is included, however, you do not show it and do not give a statement on how this was modelled.

However, these are technicalities that a staff crystallographer can take over. However, it has to be done.

Reviewer #2 (Remarks to the Author):

This paper describes interactions between polyaromatic capsule and oligo(ethylene oxide)s. The polyaromatic capsule binds a single molecule of the short oligo(ethylene oxide)s in cramming fashion. The longer oligomers are bound by the capsule in a threading fashion. The results are interesting and rather unique. Therefore, the paper has merits for publication. However, there are some points which should be made clear.

1) The reviewer is wondering what is the difference between cramming and threading. Even in the case of threading, the unit is cramming. He feels that the threading is a repeated manner of cramming. He understands that pentamer is small enough to accommodate the chain in a single capsule, but decamer is too longer for the cavity. Therefore, a part of oligo(ethylene oxide) is excluded.

2) Oligo(ethylene oxide)s seem to be highly hydrated. Therefore, much energy is required for dehydration from the oligo(ethylene oxide) chain. The reviewer wonders how the energy is compensated.

3) What would happen if oligo(ethylene oxide)s with OH end groups are used. What kinds of difference are there in the complexes of the capsule with oligo(ethylene oxide)s with methoxy end groups or OH groups.

4) Are there any other evidences for the threading structure than thermodynamic data. How about the mass spectral evidence of the isolated product.

5) The reviewer is interested in the mechanism of the formation of the complexes. Threading or clipping? The complexes appear to be reversible and dynamic. How stable are the complexes?

Reviewer #3 (Remarks to the Author):

The submitted manuscript entitled "Cramming vs. Threading: Long Amphiphilic Oligomers into a Polyaromatic Capsule" describes the complexation of oligoethylene glycols (OEG's) by a self-assembled Pt-based aromatic capsule. The authors claim the encapsulation of several shorter OEG's and interpenetrated pseudo-rotaxane style "threading" by longer oligomeric chains. These claims are more or less supported variously by ITC, NMR, MS and X-ray data.

Unfortunately, I don't recommend acceptance of the manuscript in the present form for the following reasons:

1) Novelty: The authors have published at least two other recent manuscripts (reference #15 in the submission) concerning the complexation of polar, water soluble monomers/oligomers by this aromatic capsule with similar results in the case of oligomeric lactic acids. I don't really see how this is greatly different than those reports, especially if the "pseudo-rotaxane" nature of the complexes for longer OEG's is neglected (see below). The cyclodextrin work by Harada and others (reference #10 in the submission) many years ago established the complexation of OEG's by those hydrophobic cavities in aqueous solution, so this isn't a startling

2. Support for "Threading": The complexation of the oligomers by the capsule is pretty solid from the preponderance of NMR, ITC and MS data, clinched by the X-ray structure with the 5EO oligomer. However, I don't feel the authors have sufficiently proven the pseudo-rotaxane nature of the products in the case of the longer oligomers (e.g. 10EO and OEO1000/2000) at all. The extent of the "proof" in these cases is the assertion that the oligomers can't fit entirely in the capsular interior. Fair enough, I agree. The further conclusion that the products must be threaded rather than merely containing a terminal complexation of what portion of the OEG in question fits in the cavity has not been demonstrated by the authors and remains in question to me. The sum of the authors' support for this claim is NMR line broadening of the OEG chain in all cases. This is not enough to convince me that it isn't merely the result of a dangling chain end out one of the capsule's fissures. The line broadening of the whole oligomeric chain could be a result of interactions with the interior *and the exterior* of the capsule. In fact, the authors' observations of only 2:1 complexation with the OEO1000/2000 guests actually supports solely terminal complexation of these guests by the capsules. The OEO2000 oligomer is certainly long enough to engage in 3:1 or 4:1 stoichiometries, regardless of their claim that charge repulsion limits this possibility. That sounds like a convenient justification for their results given the assembly/complexation in water. I would suggest either attempting a "capping" reaction with HO-terminated OEG's (though see 3. below) or further attempts to crystallize the 1:1 or 2:1 complexes to provide definitive evidence for threading.

3. While I appreciate that Pt based (pyridyl) ligand exchange is generally relatively slow, the authors don't provide any discussion regarding the stability or integrity of the capsule under the conditions of the experiments. This is critical to just about all of the conclusions that they make in the paper and should be explicitly addressed somewhere, if only briefly.

For the comments of Reviewer #1:

This is a very interesting and competently written manuscript that I recommend for publication after minor revision. The formation of threaded capsules on ethylenoxide-oligomers is truly fascinating and the large high field shifts in the NMR, the ITC, MS and XRD evidence is overwhelming.

We appreciate having very positive evaluation (such as *interesting, competently written, truly fascinating, and overwhelming evidence*) on this work from the Reviewer #1.

1) Are the counterions (nitrates) also within the capsule, or outside...? In the XRD in the SI, you omit those. From the MS, you say, the nitrates are in part included with the main signals, however, it is not clear to me if inside or outside the capsule.

The capsule has four nitrate ions as counterion. Whereas host-guest complex **1•5EO** also has four nitrate ions, we could not observe clear electron density for the counterions, because of low quality of the obtained crystals. However, the crystal structure indicates that all of the counterions exist outside the capsule due to the lack of remaining space in the capsule's cavity. The majority of the observed MS peaks are derived from host-guest complexes attaching nitrate ions outside the capsule.

2) What is the effect on encapsulation on the ^{195}Pt -NMR...?

According to the reviewer's question, we first examined the ^{195}Pt NMR analysis of capsule **1**. As a control experiment, we could observe a broad signal at -2606 ppm for $(\text{NH}_3)_4\text{PtCl}_2$ in D_2O . However, ^{195}Pt NMR spectrum (107 Hz, r.t.) of capsule **1** showed no peaks in the range of -2000 to -3000 ppm (even spending 6 h for the measurement time), most probably due to the poor solubility of our capsule (<0.8 mM) in D_2O . Unfortunately, we could not discuss the requested effect in this sample. In the crystal structure of **1•5EO** (Fig. 3c), the closest distance between the Pt(II) center and the guest's oxygen atom is 3.9 \AA , which suggests no specific interaction between the Pt(II) center and the guest.

3) In both crystal structures you do have A- and B-alerts in the CIFcheck files. However, you do not give any explanation for them. In addition, the second structure does not qualify for regular uptake in the CCDC, as R1 and wR2 values are high ($R1 > 12\%$). I am familiar with such type of structures, so I do believe, the structures may be acceptable for publication.

However, at least every effort should be made to prove that the current state is as good as it gets. Please include a competent statement in the supporting information. I realize, that the thermal ellipsoids of the dangling residues are large (as expected). You even state that solvent is included, however, you do not show it and do not give a statement on how this was modelled.

We have already described appropriate statements for the A-alert messages in the CIF files. According to the reviewer's comments, we added the following explanations for the A- and B-alert messages on the checkCIF files of **1** and **1•5EO** to the revised SI (page 37-40).

Refinement details of X-ray crystallographic analysis of **1 (SK287)**

The crystal structure of capsule **1** was solved using SHELXT (Sheldrick, 2015) and then refined with SHELXL (Sheldrick, 2015). Carbon-bound hydrogen atoms were included in idealized positions and refined using a riding model. Disorder atoms were modelled using standard crystallographic methods including constraints, restraints, and rigid bodies. The contribution of the electron density associated with greatly disordered counterions ($4\cdot\text{NO}_3^-$) and solvent molecules (H_2O), which could not be modelled with discrete atomic positions, were handled using the solvent mask in the Olex² program. There are several

error messages (level A and B) in the checkCIF, because of the following reasons. Nevertheless, the quality of the data ($R_1 = 0.0834$, $wR_2 = 0.2407$) is more than sufficient to establish the connectivity of capsule **1**.

PLAT029_ALERT_3_A

PROBLEM: _diffn_measured_fraction_theta_full value Low . 0.855 Why?

RESPONSE: This alert is generated because crystal was severely damaged during the X-ray irradiation. Diffraction intensity in the images of the last part of the schedule was dramatically decayed and the images were not used for the structure refinement.

PLAT413_ALERT_2_A

PROBLEM: Short Inter XH3 .. XHn H46A ..H53I 1.88 Ang.

RESPONSE: This alert is generated because there is a large amount of disorder in the structure. In particular, the disordered side chains are very dynamic and may be considered as a solvent. Short contacts between disordered fragments are to be expected.

PLAT220_ALERT_2_B

PROBLEM: Non-Solvent Resd 1 C Ueq(max)/Ueq(min) Range 8.2 Ratio

RESPONSE: This alert is generated because there is a large amount of disorder in the structure. In particular, the disordered side chains are very dynamic and may be considered as a solvent, which has considerably large Ueq.

PLAT234_ALERT_4_B

PROBLEM: Large Hirshfeld Difference O2C --C47C 0.30 Ang.

PROBLEM: Large Hirshfeld Difference O6C --C53C 0.30 Ang.

PROBLEM: Large Hirshfeld Difference C45B --C46B 0.28 Ang.

RESPONSE: These alerts are generated because there is a large amount of disorder in the structure. In particular, the disordered side chains are very dynamic and may be considered as a solvent. A disorder model for the side chains could not be built because of the tight disorder.

PLAT241_ALERT_2_B

PROBLEM: High 'MainMol' Ueq as Compared to Neighbors of O4A Check

PROBLEM: High 'MainMol' Ueq as Compared to Neighbors of C48B Check

RESPONSE: These alerts are generated because there is a large amount of disorder in the structure. In particular, the disordered side chains are very dynamic and may be considered as a solvent. A disorder model for the side chains could not be built because of the tight disorder.

Refinement details of X-ray crystallographic analysis of 1•5EO (SK291)

The crystal structure of host-guest complex **1•5EO** was solved using SHELXT (Sheldrick, 2015) and then refined with SHELXL (Sheldrick, 2015). Carbon-bound hydrogen atoms were included in idealized positions and refined using a riding model. Disorder atoms were modelled using standard crystallographic methods including constraints, restraints, and rigid bodies. The contribution of the electron density associated with greatly disordered counterions ($4 \bullet \text{NO}_3^-$) and solvent molecules (H_2O), which could not be modelled with discrete atomic positions, were handled using the solvent mask in the Olex² program. There are several error messages (level A and B) in the checkCIF, because of the following reasons. Nevertheless, the quality of the data is more than sufficient to establish the connectivity of the capsule and penta(ethylene oxide) structures.

THETM01_ALERT_3_A

PROBLEM: The value of $\sin(\theta_{\max})/\lambda$ is less than 0.550

Calculated $\sin(\theta_{\max})/\lambda = 0.5000$

RESPONSE: Refinement was performed using the reflection data of 1.0 Å resolution, since the values of Rint and mean $F^2/\sigma(F^2)$ in the resolution shell between 1.04 and 1.00 Å were 75.5% and 0.23, respectively. Considerably large R values might be due to a poor quality of the crystal and reduced number of parameters used for the refinement. The disordered solvent (water) molecules and nitrate anions could not be modeled and treated with solvent mask of Olex2 program.

PLAT412_ALERT_2_A

PROBLEM: Short Intra XH3 .. XHn H32C ..H50G 1.48 Ang.

PROBLEM: Short Intra XH3 .. XHn H50L ..H52E 1.49 Ang.

RESPONSE: These alerts are generated because there is a large amount of disorder in the structure. In particular, the disordered side-chains are very dynamic and may be considered as a solvent. Short contacts between disordered fragments are to be expected.

RINTA01_ALERT_3_B

PROBLEM: The value of Rint is greater than 0.18

Rint given 0.198

RESPONSE: Considerably large Rint values is due to a poor quality of the crystal and the severely disordered side chains, counterions, and solvents, resulting in weak diffraction at the high angle region.

PLAT020_ALERT_3_B

PROBLEM: The Value of Rint is Greater Than 0.12 0.198 Report

RESPONSE: Considerably large Rint values is due to a poor quality of the crystal and the severely disordered side chains, counterions, and solvents, resulting in weak diffraction at the high angle region.

PLAT026_ALERT_3_B

PROBLEM: Ratio Observed / Unique Reflections (too) Low .. 33% Check

RESPONSE: This alert is generated because there is a large amount of disorder in the structure. This resulted in the weak intensity of the diffraction, where $F^2/\sigma(F^2)$ falls below 2.0 in the resolution shell between 1.36 and 1.26 Å.

PLAT084_ALERT_3_B

PROBLEM: High wR2 Value (i.e. > 0.25) 0.40 Report

RESPONSE: Considerably large wR values might be due to a poor quality of the crystal and reduced number of parameters used for the refinement. Refinement was performed using the reflection data of 1.0 Å resolution, since the values of Rint and mean $F^2/\sigma(F^2)$ in the resolution shell between 1.04 and 1.00 Å were 75.5% and 0.23, respectively.

PLAT213_ALERT_2_B

PROBLEM: Atom N1B has ADP max/min Ratio 4.1 prolat

RESPONSE: This alert is generated because there is a large amount of disorder in the structure. A disorder model could not be built because of the tight disorder.

PLAT234_ALERT_4_B

PROBLEM: Large Hirshfeld Difference Pt2 --N2A 0.26 Ang.
 PROBLEM: Large Hirshfeld Difference N1C --C1C 0.26 Ang.
 PROBLEM: Large Hirshfeld Difference C2C --C3C 0.26 Ang.
 PROBLEM: Large Hirshfeld Difference C16D --C17D 0.26 Ang.
 PROBLEM: Large Hirshfeld Difference C17C --C18C 0.26 Ang.
 PROBLEM: Large Hirshfeld Difference C22B --C23B 0.28 Ang.
 PROBLEM: Large Hirshfeld Difference C24B --C26B 0.28 Ang.
 PROBLEM: Large Hirshfeld Difference C27B --C32B 0.28 Ang.
 PROBLEM: Large Hirshfeld Difference C28B --C29B 0.28 Ang.
 PROBLEM: Large Hirshfeld Difference C29B --C30B 0.26 Ang.
 PROBLEM: Large Hirshfeld Difference C37B --C38B 0.30 Ang.

RESPONSE: These alerts are generated because there is a large amount of disorder in the structure. In particular, the disordered side chains are very dynamic and may be considered as a solvent. A disorder model for the side chains could not be built because of the tight disorder.

PLAT241_ALERT_2_B

PROBLEM: High 'MainMol' Ueq as Compared to Neighbors of C48C Check
 PROBLEM: Low 'MainMol' Ueq as Compared to Neighbors of O4D Check
 PROBLEM: Low 'MainMol' Ueq as Compared to Neighbors of C48A Check

RESPONSE: These alerts are generated because there is a large amount of disorder in the structure. In particular, the disordered side chains are very dynamic and may be considered as a solvent. A disorder model for the side chains could not be built because of the tight disorder.

PLAT342_ALERT_3_B

PROBLEM: Low Bond Precision on C-C Bonds 0.0416 Ang.

RESPONSE: Considerably large wR values might be due to a poor quality of the crystal and reduced number of parameters used for the refinement. Refinement was performed using the reflection data of 1.0 Å resolution, since the values of Rint and mean $F^2/\sigma(F^2)$ in the resolution shell between 1.04 and 1.00 Å were 75.5% and 0.23, respectively.

PLAT369_ALERT_2_B

PROBLEM: Long C(sp2)-C(sp2) Bond C24B - C26B . 1.59 Ang.
 PROBLEM: Long C(sp2)-C(sp2) Bond C39A - C43A . 1.58 Ang.

RESPONSE: These alerts are generated because there is a large amount of disorder in the structure. The positions of concerned carbon atoms are averaged one of disordered structure. A disorder model for the side chains could not be built because of the tight disorder.

PLAT412_ALERT_2_B

PROBLEM: Short Intra XH3 .. XHn H45F .. H47H 1.76 Ang.

RESPONSE: This alert is generated because there is a large amount of disorder in the structure. In particular, the disordered side chains are very dynamic and may be considered as a solvent. Short contacts between disordered fragments are to be expected.

For the comments of Reviewer #2:

This paper describes interactions between polyaromatic capsule and oligo(ethylene oxide)s. The polyaromatic capsule binds a single molecule of the short oligo(ethylene oxide)s in cramming fashion. The longer oligomers are bound by the capsule in a threading fashion. The results are interesting and rather unique. Therefore, the paper has merits for publication.

We again appreciate having very positive evaluation (such as *interesting and rather unique*) on this work from the Reviewer #2.

1) The reviewer is wondering what is the difference between cramming and threading. Even in the case of threading, the unit is cramming. He feels that the threading is a repeated manner of cramming. He understands that pentamer is small enough to accommodate the chain in a single capsule, but decamer is too longer for the cavity. Therefore, a part of oligo(ethylene oxide) is excluded.

To clarify the structural difference between cramming and threading, we added the following sentence to the introduction part (page 2): “In contrast to such common complexation behavior, herein we report that a molecular capsule binds one molecule of linear amphiphilic oligomers in two different manners, a cramming or threading fashion, depending on the chain length. *Namely, the shorter oligomers are fully accommodated in the closed cavity, yet the longer ones stick the ends out from the cavity (Fig. 1a)*”.

2) Oligo(ethylene oxide)s seem to be highly hydrated. Therefore, much energy is required for dehydration from the oligo(ethylene oxide) chain. The reviewer wonders how the energy is compensated.

In this work, detailed isothermal titration calorimetry (ITC) studies and X-ray crystallographic analysis revealed that *enthalpic stabilization is a main driving force for the host-guest formations through multiple CH- π and hydrogen bonding interactions*. We have already mentioned the related discussions on page 5 in the main text.

3) What would happen if oligo(ethylene oxide)s with OH end groups are used. What kinds of difference are there in the complexes of the capsule with oligo(ethylene oxide)s with methoxy end groups or OH groups.

The capsule can bind such oligomers (*i.e.*, **5EO'** and **8EO'** shown in Fig. 1c) in a manner similar to methyl-capped **5EO** and **8EO**. “*The terminal methyl groups on OEOs are not essential for the formation of complexes **1•nEO**. For example, non-capped pentamer **5EO'** and octamer **8EO'** (Fig. 1c) were quantitatively encapsulated within **1** under the same conditions, respectively*” has been already described on page 4 in the main text.

4) Are there any other evidences for the threading structure than thermodynamic data. How about the mass spectral evidence of the isolated product.

We have fortunately obtained a clear MS evidence for threading-type host-guest complex **1•10EO**. “*The product is stable enough under standard ESI-TOF MS conditions so that the host-guest ratio could be unambiguously confirmed by the MS analysis: prominent molecular ion peaks were observed at m/z 2113.4, 1388.3, and 1025.7 for $[1•10EO - n•NO_3^-]^{n+}$ ($n = 2, 3, \text{ and } 4, \text{ respectively; Fig. 5b}$)*” has been already described in the main text (page 6).

5) The reviewer is interested in the mechanism of the formation of the complexes. Threading or clipping? The complexes appear to be reversible and dynamic. How stable are the complexes?

This is a very important question. We employed robust Pt(II)-linked capsule **1**, instead of the Pd(II) analogue, in this work. As a response to this question, we added the following sentence to the main text (page 4): “*The coordination bond between the Pt(II) ion and the bipyridine ligand is inert at ambient temperature so that the encapsulation of the linear as well as cyclic oligomers by **1** should occur without any dissociation step of the capsular frameworks*”.

For the comments of Reviewer #3:

The submitted manuscript entitled "Cramming vs. Threading: Long Amphiphilic Oligomers into a Polyaromatic Capsule" describes the complexation of oligoethylene glycols (OEG's) by a self-assembled Pt-based aromatic capsule. The authors claim the encapsulation of several shorter OEG's and interpenetrated pseudo-rotaxane style "threading" by longer oligomeric chains. These claims are more or less supported variously by ITC, NMR, MS and X-ray data. Unfortunately, I don't recommend acceptance of the manuscript in the present form for the following reasons:

We would like to emphasize the novelty of this work and the structural characterization of the threading products by the following responses to the comments of Reviewer #3.

1) Novelty: The authors have published at least two other recent manuscripts (reference #15 in the submission) concerning the complexation of polar, water soluble monomers/oligomers by this aromatic capsule with similar results in the case of oligomeric lactic acids. I don't really see how this is greatly different than those reports, especially if the "pseudo-rotaxane" nature of the complexes for longer OEG's is neglected (see below). The cyclodextrin work by Harada and others (reference #10 in the submission) many years ago established the complexation of OEG's by those hydrophobic cavities in aqueous solution, so this isn't a startling

We would like to mention the following two points as the novelties of this work.

(i) We have already reported the encapsulation of various *hydrophobic* and *hydrophilic* guests by capsule **1**. *The sizes of the previous guests are smaller than that of the capsule cavity.* These host-guest behaviors are common and quite similar to that of other capsules reported by, e.g., the Rebek and Gibb groups (ref. 3-8). On the other hand, in this work, we for the first time succeeded in the binding of long oligomer guests by the capsule. The great difference is that the present guest size is much larger than the cavity size of the capsule, which has been emphasized in the introduction part. The resultant threading complexation, even though the terminal complexation, is quite unusual for molecular capsules. The key of this achievement is caused by the judicious choice of long *amphiphilic* guests and a *polyaromatic* capsular host.

(ii) We agree that the complexation of oligo(ethylene glycol)s by α -CD and the derivatives has been intensively studied since the first report by the Harada group in 1990 (ref. 23). However, to our surprise, except for the CD-based hosts, there is no report on molecular hosts capable of binding oligo(ethylene glycol)s in solution. The CDs are *tubular* hosts composed of *covalent*, *neutral*, and *aliphatic* frameworks with *open* cavities. In contrast, capsule **1** is a capsular host composed of *coordination bonded*, *cationic*, and *polyaromatic* frameworks with a *closed* cavity. Their structural characters are completely different.

Therefore, we believe that the present work is not simple extensions of previous our and CDs works but a discovery about unique host-guest complexations with wide-ranging amphiphilic oligomers (M.W. = approximately 200-2000) in water.

2. Support for "Threading": The complexation of the oligomers by the capsule is pretty solid from the preponderance of NMR, ITC and MS data, clinched by the X-ray structure with the 5EO oligomer.

We appreciate having positive evaluation on our structural analyses of "the complexation of the oligomers by the capsule" from the Reviewer #3. The *composition* (1:1 or 2:1) of the host-guest structures was unambiguously confirmed by NMR, ITC, MS, and X-ray crystal

(for **1•5EO**) analyses.

However, I don't feel the authors have sufficiently proven the pseudo-rotaxane nature of the products in the case of the longer oligomers (e.g. **10EO** and **OEO1000/2000**) at all. The extent of the "proof" in these cases is the assertion that the oligomers can't fit entirely in the capsular interior. Fair enough, I agree. The further conclusion that the products must be threaded rather than merely containing a terminal complexation of what portion of the OEG in question fits in the cavity has not been demonstrated by the authors and remains in question to me. The sum of the authors' support for this claim is NMR line broadening of the OEG chain in all cases. This is not enough to convince me that it isn't merely the result of a dangling chain end out one of the capsule's fissures. The line broadening of the whole oligomeric chain could be a result of interactions with the interior *and the exterior* of the capsule. In fact, the authors' observations of only 2:1 complexation with the **OEO1000/2000** guests actually supports solely terminal complexation of these guests by the capsules.

We understood that, the reviewer suggests that the *terminal* complexation of the oligomers with the capsules occurs, rather than the *threading* complexation, in the case of long oligomers such as **10EO** and **OEO^{1000/2000}**. The terminal complexation is a key step for the formation of the thermodynamically most stable product. However, we would like to definitely conclude the selective formation of the threading host-guest complexes in this work, on the basis of the following three experimental data.

(i) We observed significant ¹H NMR line broadening of the guests bound by the capsule only in the case of the long oligomers. As the reviewer mentioned, we can not explain that the line broadening stems from the formation of threading host-guest complexes. However, the *upfield shifts* of the bound guest signals indicate the formation of the threading complexes. The proton signals of *free* **10EO** in D₂O are observed at 3.4-3.8 ppm. On the other hand, after the complex formation, we observed the broad proton signals of **10EO** in the range of 1.1 to -0.2 ppm (Fig. 5c).

To clarify this finding, "Large upfield shifts of the guest ethylene signals indicate the incorporation of the middle part of the guest chain into the polyaromatic cavity" was added to the revised manuscript (page 6). In addition, we added the following scheme to Supplementary Fig. 31d. It is reasonable that the upfield shifts are slightly weakened in the case of much longer oligomer **OEO¹⁰⁰⁰** (1.8 to -0.2 ppm; Fig. 6b).

(ii) In ^1H NMR spectra, the terminal methyl signals for bound **10EO** and **OEO^{1000/2000}** are significant broadened even at various temperatures, in sharp contrast to those for bound **5EO**. These findings are not suitable to assign the selective encapsulation of the terminal part(s) of the long oligomers. In contrast, host-guest complex **1•9EO** displayed both thoroughly broad signals and relatively sharp signals derived from bound **9EO** in the highly upfield region (1.8-0.2 ppm in Supplementary Fig. 32) at room temperature. This unusual signal pattern indicates that one side of the long **9EO** chain partially sticks out of the capsular shell, which might be assigned as a *terminal* complexation. We modified the related sentence in page 6 and added the following scheme to Supplementary Fig. 33d.

(iii) Since the capsule provides twelve methoxyethoxy groups as hydrophilic substituents on the outer surface, there is no space for the polyaromatic frameworks of capsule **1** to interact with the oligomers from the *outside*. In addition, we observed only *sharp* ^1H NMR signals derived from the *free oligomers at a typical aliphatic region*, when an excess amount of the guest was added to the capsule solution in D_2O .

Therefore, we added “*Due to the twelve, exterior hydrophilic groups, the polyaromatic framework can interact with guest molecules only from the inside*” to the introduction part (page 3), in response to the following referee’s comment: “The line broadening of the whole oligomeric chain could be a result of interactions with the interior *and the exterior* of the capsule”.

The OEO2000 oligomer is certainly long enough to engage in 3:1 or 4:1 stoichiometries, regardless of their claim that charge repulsion limits this possibility. That sounds like a convenient justification for their results given the assembly/complexation in water.

In the Harada’s works, the complexation of α -CD with oligo(ethylene glycol)s (M.W. > 1000) generates abacus-type host-guest structures because of multiple hydrogen-bonding interactions between the CD hosts. On the other hand, the present capsule provides repulsive positive charges (4+) and, with the lack of attractive interactions between the capsules, 3:1 and 4:1 host-guest complexes are entropically less favorable than the 2:1 complex. The $\Delta\Delta S$ value for the formation of $(1)_2\cdot\text{OEO}^{1000}$ (-99.6 kJ/mol) is much lower than that of **1•10EO** (-25.1 kJ/mol), as confirmed by the ITC experiment (Table 1), so that entropic energy costs for the formation of the 3:1 and 4:1 host-guest complexes should be quite high in this system.

To clarify this point, we added “The observed, unusual regulation of the number of the threaded capsules by the long chains (up to ~ 44 -mer) can be explained by the electrostatic repulsion between the tetravalent capsules, *besides the energetic balance between the enthalpic gains and the entropic losses*” to the revised manuscript (page 8).

I would suggest either attempting a "capping" reaction with HO-terminated OEG's (though see 3. below) or further attempts to crystallize the 1:1 or 2:1 complexes to provide definitive evidence for threading.

In this work, we focused methyl-capped oligo(ethylene oxide)s to reveal the host-guest structures in a cramming or threading fashion. Exploration of capping reactions using long oligomers without methyl caps is our next project. We have to carefully choose bulky and hydrophilic molecules as the caps and also efficient ligation reactions usable in water under mild conditions, which is beyond the scope of this work. We could not obtain the suitable crystals of 2:1 host-guest complexes even after spending more than one year under various conditions, most probably due to the flexible long chains partially sticking out of the capsule framework.

3. While I appreciate that Pt based (pyridyl) ligand exchange is generally relatively slow, the authors don't provide any discussion regarding the stability or integrity of the capsule under the conditions of the experiments. This is critical to just about all of the conclusions that they make in the paper and should be explicitly addressed somewhere, if only briefly.

We thank this helpful suggestion from the Reviewer #3. In this work, we employed robust Pt(II)-linked coordination capsule **1** to simplify the interactions between the capsule and long amphiphilic oligomers.

According to the reviewer's suggestion, we added "*The coordination bond between the Pt(II) ion and the bispyridine ligand is inert at ambient temperature so that the encapsulation of the linear as well as cyclic oligomers by **1** should occur without any dissociation step of the capsular frameworks*" to the main text (page 4).

REVIEWERS' COMMENTS:

Reviewer #1 (Remarks to the Author):

In the revised version the authors have addressed all my scientific issues. It is unfortunate that no Pt-NMR Signal was observed, but this happens especially with these low concentrations.

No further issues.

Reviewer #2 (Remarks to the Author):

The reviewer has now understood what the authors answered. He thinks that the paper is publishable for the Journal.

Reviewer #3 (Remarks to the Author):

Regarding the authors' responses to my concerns regarding our manuscript:

1) I do appreciate that this host is much different than Cyclodextrins examined by Harada. In that vein I suppose the novelty is present, though I consider it at the edge of being "new" enough for this venue.

2) I remain unconvinced regarding the threading character of the complexes. It is tenuous evidence to rely on the qualitative broadness of NMR signals and small differences in their upfield shifts as "proof" of their pseudorotaxane geometries, rather than terminal complexation. However, without a fair bit of extra synthetic work there is little extra the authors can provide in the current context.

I do dispute the assertion that there is no room on the outer shell of the capsule for at least some interaction with the ethylene glycol guests. However, that will have to remain in contention in the absence of some quantitative evaluation provided by the authors.

3) I agree that there is likely little or no dissociation of the capsule under the conditions of the experiment. One does wonder if that is true at 75 deg C though...

I, somewhat begrudgingly, am ok with recommending publication of the manuscript in Nature Communications.

For the comments of Reviewer #3:

1) I do appreciate that this host is much different than Cyclodextrins examined by Harada. In that vein I suppose the novelty is present, though I consider it at the edge of being "new" enough for this venue.

We appreciate having very positive evaluation on our work from the viewpoint of the novelty.

2) I remain unconvinced regarding the threading character of the complexes. It is tenuous evidence to rely on the qualitative broadness of NMR signals and small differences in their upfield shifts as "proof" of their pseudorotaxane geometries, rather than terminal complexation. However, without a fair bit of extra synthetic work there is little extra the authors can provide in the current context. I do dispute the assertion that there is no room on the outer shell of the capsule for at least some interaction with the ethylene glycol guests. However, that will have to remain in contention in the absence of some quantitative evaluation provided by the authors.

According to the claim from the third reviewer, we added a host-guest structure with terminal complexation as a possible equilibrium mixture to Figure 5a, whereas we have already added the similar scheme to Supplementary Fig. 31b. "... *small differences in their upfield shifts ...*" is wrong. The ¹H NMR spectral patterns of (i) the cramming complexation (e.g., Fig. 2b,c), (ii) the threading complexation (e.g., Fig. 5c), and the terminal complexation (e.g., Supplementary Fig. 32) are clearly different. We will study the details of the capping as well as connecting reactions of the guest threading into the host in our next project from the viewpoint of polymer and materials chemistry. Besides the observation of sharp ¹H NMR signals derived from the excess guests at a typical aliphatic region (e.g., Supplementary Fig. 30), the ITC analysis of **1•10EO** showed a single inflection point at a guest's molar ratio of 1.0 in the range of 0.3 to 1.8 (Figure 4c). This result quantitatively reveals the formation of only 1:1 host-guest structure without the formation of other host-guest complexes, e.g., by the external interactions of the host with the excess guests.

3) I agree that there is likely little or no dissociation of the capsule under the conditions of the experiment. One does wonder if that is true at 75 deg C though...

We have already examined the temperature-dependent NMR analysis of **1•10EO** in D₂O. The ¹H NMR spectrum indicates that the host-guest structure remains intact even at 75 °C (Supplementary Fig. 31a). We observed upfield-shifted signals for bound **10EO** without signals assignable to free **1** in the spectrum.

I, somewhat begrudgingly, am ok with recommending publication of the manuscript in Nature Communications.

In addition to the first and second reviewers, we sincerely appreciate having the recommendation from the third reviewer for publication of our manuscript in Nature Communications.